# An Experimental Brackish Aquaponic System Using Juvenile Gilthead Sea Bream *(Sparus aurata)* and Rock Samphire (*Crithmum maritimum*)

**Nikolaos Vlahos** [1,2,*], **Efi Levizou** [3], **Paraskevi Stathopoulou** [1], **Panagiotis Berillis** [1], **Efthimia Antonopoulou** [4], **Vlasoula Bekiari** [2], **Nikos Krigas** [5], **Konstantinos Kormas** [1] and **Eleni Mente** [1]

1    Department of Ichthyology and Aquatic Environment, School of Agricultural Sciences, University of Thessaly, 38446 Fytoko Volos, Magnesia, Greece
2    Department of Animal Production, Fisheries and Aquaculture, School of Agricultural Sciences, University of Patras, 30200 Mesolonghi, Greece
3    Department of Agriculture Crop Production and Rural Environment, School of Agricultural Sciences Fytoko, University of Thessaly, 38446 Fytoko Volos, Magnesia, Greece
4    Laboratory of Animal Physiology, Department of Zoology, School of Biology, Aristotle University of Thessaloniki, 54124 Thessaloniki, Greece
5    Institute of Plant Breeding and Genetic Resources, Hellenic Agricultural Organization, Demeter, Thermi, 57001 Thessaloniki, Greece
*    Correspondence: nvlahos@uth.gr; Tel.: +30-24210-93176

**Abstract:** Brackish aquaponics using Mediterranean fish and plants provides an alternative opportunity for a combined production of high-quality food products with high commercial and nutritional value. This is the first study that investigates the effect of two different salinities (8 and 20 ppt) on growth and survival of *Sparus aurata* and *Crithmum maritimum* along with the cellular stress pathways using the activation of heat shock proteins (HSPs) and mitogen-activated protein kinase (MAPK) protein family members and the water bacterial abundance. In total, 156 fish were used (average initial weight of 2.55 g, length of 5.57 cm) and 36 plants (average initial height of 8.23 cm) in floating racks above the 135 L fish tanks. Survival rate for both organisms was 100%. *C. crithmum* grew better at 8 ppt (t-test, $p < 0.05$). The growth rate of *S. aurata* was similar for both treatments ($p > 0.05$). HSPs and MAPK were differentially expressed, showing tissue-specific responses. The average bacterial abundance at the end of the experiment was higher ($p < 0.05$) in the 20 ppt ($18.6 \pm 0.91$ cells $\times 10^5$/mL) compared to the 8 ppt ($6.8 \pm 1.9$ cells $\times 10^5$/mL). The results suggest that the combined culture of euryhaline fish and halophytes provides good quality products in brackish aquaponics systems.

**Keywords:** aquaponics; brackish water; gilt-head juvenile sea bream; rock samphire; water reuse; HSPs; MAPKs

## 1. Introduction

Agricultural production systems are being affected by unsustainable management practices and changing climatic conditions. To respond to this situation, designing strategies to ensure food security for all represents a continual challenge. Therefore, research on alternative food production methods to develop novel ideas [1] for reducing the environmental load by using innovative and alternative food production systems, such as aquaponic systems, is necessary. Aquaponics is a modern co-cultivation method that combines fish farming and plant cultivation in a recycled system and uses fish waste for

plant nutrition [2]. It is an environmentally friendly food culture method that combines aquaculture and hydroponics with or without the limited use of chemical fertilizers for the plants [3].

Hydroponic systems operate based on natural biological processes, mainly nitrification [4]. In aquaponics, fish waste and uneaten food are used as fertilizer and provide valuable nutrients to promote plant growth. In this transformation, the role of nitrifying bacteria is important [5]. The function of an aquaponic system is based on the biochemical processes that take place in the biological filter, particularly the biochemical oxidation of ammonia to nitrite and nitrate through the autotrophic bacteria *Nitrosomonas* sp. and *Nitrobacter* sp., respectively. Nitrate is not toxic for fish and is useful for plants [6,7], which use it as a nutrient (adsorption process) [5,8]. Water quality and all of the above parameters have an impact on fish and plant growth in aquaponic systems [9]. The water directly flows to the plants by a pump and is supplemented with nitrate ions and other nutrients (potassium, iron, etc.) as fertilizers that are absorbed by the plant roots [3]. At the end, the water is transferred back to the fish tanks, and it is up to 97% free of nitrates [3]. Water quality is a primary environmental consideration for optimizing the production processes of aquaponic systems, and it has a direct impact on the health of fish and plants by purifying them to that they reach their optimum welfare potential [9].

Assessing the performance of a biofilter in a recirculating aquaculture system (RAS), especially in an aquaponic system, is difficult due to the many and interacting factors that need to be evaluated. To analyse the biochemical process of the filter bed, it is essential to control the water parameters [10]. The total ammonium nitrogen (TAN), nitrite ($NO_2^-$), nitrate ($NO_3^-$), pH, salinity and dissolved oxygen (DO) are among the most important parameters that need to be measured daily. The biofilter provides a large surface area and proper temperature, pH and dissolved oxygen to accommodate the bacteria and keep the system healthy. The carrying capacity of the aquaponic system may be influenced by several other factors, such as the filter bed, filter media, nutrient input and nitrification rate [11]. The nutrient inputs are divided into two main categories associated with the effect of the nitrification process on bacterial activity. The main sources of inorganic nutrients are nitrogen, phosphorus or trace elements, while organic compounds in the water provide the carbon source for all heterotrophic microorganisms. Bacteria require energy from the conversion of large amounts of nitrogen that is oxidized to nitrate. Bacteria also consume organic carbon for their cellular synthesis [12]. The nutrients, C/N ratio, feed composition, bacteria abundance, filter capacity, water quality (pH, DO, alkalinity and temperature), disinfection and hydraulic retention time are the most relevant factors that affect the carrying capacity of aquaponic systems [3–5,11]. Moreover, the water quality can disturb the homeostasis of aquatic organisms, including fishes. Teleosts can adapt to both external and internal changes associated with intracellular members of heat shock proteins (HSPs) (e.g., temperature [13], anoxia [14], salinity [15]). Furthermore, the activation of mitogen-activated protein kinases (MAPKs) plays a critical role in signal transduction and physiological functions (e.g., [16]). Thus, members of both MAPKs and HSPs are often used as potential biomarkers for freshwater biomonitoring (e.g., [17]). Moreover, in gilthead seabream, both HSPs and MAPKs are detectable, even during early embryonic stages [18].

Freshwater aquaponics is among the most widespread technical practices developed for agriculture. More than 150 different plant species have been successfully used in aquaponics. Lettuce, tomato, basil, eggplant, pepper and spinach are among the most common species cultivated in aquaponic systems [19–21]. The limited freshwater volume for aquaculture and agriculture, including the gradual increase of saline soils [22] in combination with an estimated >50% of groundwater sources containing brackish water, have stimulated the production of food by the use of saline soils or irrigation water [23]. Thus, the use of brackish water in aquaponic systems that include plants that are naturally adapted to salinity (halophytes) is very important [24]. Aquaponics with brackish water may combine the rearing Mediterranean euryhaline species of fish, such as juvenile sea bream or sea bass, in combination with a wide variety of Mediterranean plant organisms, such as algae [23] or halophytes (e.g., rock samphire), which exhibit high commercial and/or nutritional value [5,25]. Published research on freshwater aquaponic systems is limited to only a few studies [5,22,25,26]. In addition, published research data

available for sea water aquaponics are scarce, while the corresponding studies in brackish aquaponics are minimal [5,24].

This study aims to investigate the effect of salinity on the growth performance of gilthead juvenile sea bream and rock samphire in a brackish aquaponic system and the associated water bacterial abundance under two different salinities (8 ppt and 20 ppt). The present study fills the gap on such systems in the literature and can motivate the further development of aquaponic systems using brackish water.

## 2. Materials and Methods

### 2.1. Plants and Fish

The experiment was conducted during July to September 2018, using fish obtained from a nursery facility and plants propagated at the Institute of Plant Breeding and Genetic Resources, Hellenic Agricultural Organization Demeter. In total, 156 individuals of gilthead juvenile sea breams (26 individual/system), *Sparus aurata*, with an average body weight of 2.55 ± 0.53 g and an average body length of 5.57 ± 0.33 cm were used, together with 36 rock samphire plants (*Crithmum maritimum*), with an average initial height of 8.23 ± 0.34 cm (6 individuals/system). Fish and plants were divided into two treatments under two different salinities 8 ppt and 20 ppt, respectively. All experimental procedures were conducted according to the guidelines by the EU Directive 2010/63/EU regarding the protection of animals used for scientific purposes and were applied by FELASA accredited scientists (functions A–D). The two salinities were selected for the following two reasons: (a) 8 ppt is the optimal salinity for rock samphire growth performance [27,28] in hydroponic facilities and (b) 20 ppt is the maximum salinity for the survival and growth for rock samphire.

### 2.2. Experimental Set-Up and Operation

In total, six autonomous aquaponic systems with a total volume of 135 L per system were constructed. Each treatment consisted of three rectangular fish tanks (30 cm × 60 cm × 30 cm), three hydroponic cultivation tanks as described by Somerville et al. and Nozzi et al. [3,25] as a raft method cultivation, thus, creating a volume of 54 L for the fish and a cultivation area of 1800 cm$^2$ for the plants. Each aquaponic system was supported by a biological sump filter (30 cm × 30 cm × 30 cm) a total volume of 27 L, which had a significant contribution in the nitrification process by increasing filter efficiency. In addition, the salinity of each aquaponic system was gradually decreased during a one-month period, by reducing 5 units of salinity once a week until it was stabilized from 35 ppt to 20 ppt and 8 ppt, respectively. Juvenile gilhead sea bream were acclimatized accordingly at both salinity treatments. A similar acclimatization process was followed for the plants in order to avoid osmotic shock. Using tap water of almost zero salinity, a daily stepwise increase of 50 mM NaCl was applied in the irrigation water until its salinity reached 8 ppt and 20 ppt.

A sump filter was divided into three sections, with most of the filter covered with suitable media that provided a specific surface area (SSA) for nitrifying bacteria to colonize. The mechanical filter consisted of a 1 cm mesh perforated basket that covered a surface area of 455 cm$^2$. A thick layer of sponge was added to retain the solid residues from the fish tank (uneaten food and faeces). The biofilter was fixed by a mixed media of 2–4 L bioballs, 2 L of a ceramic ring (siporax) and 2 L of lava grain with an average size of 35 mm, and it covered a surface area of 429 cm$^2$. A pump (SUNSUN, 22 W, 1000 L/h, 0.55 kg) was placed in the last part of the filter to supply the aquaponic system with water through the filter. In each system, a 400 watt lamp (Sylvania, 400 W, high pressure sodium) was placed at a distance of 60 cm from the surface of the grow beds to ensure the appropriate exposure of plants to light. A 14 h light, 10 h dark photoperiod (summer photoperiod) was set up. An air-lift pump was used to recycle the water through a filter bed during the experiments, and it had an adjusted flow of 1496 cm$^3$/min, thus creating a filtration speed of 2.24 cm/min. The oxygen levels fluctuated between

60% and 80% saturation. The system was arranged in such a way that water flowed via gravity from the hydroponic tank to the fish tank and then into the sump filter [3].

Fish were hand-fed 5% of their body weight with a commercial floating pellet diet (54% protein and 18% crude fat, BIOMAR aquafeed company, inicio pellets 1.5 mm) three times per day (at 10:00, 14:00, and 18:00) for a period of 102 days. The feeding rate was adjusted to fish weight every two weeks. Fish tanks were cleaned, and uneaten food was removed every day by siphoning. At the end of the experiment, fish were anesthetized with an 0.20 mg/L MS 222, and final fish body weights and lengths were measured.

Rock samphire plants were planted directly into the hydroponic growing beds which consisted of floating sheets of polystyrene and were fixed at net pots for plant support. The plants were placed in the clay pebbles (8–16 mm) substrate of the hydroponic bedata distance of 7 cm from one another (spacing among the central axes of the plants). The homogeneity of photosynthetically active radiation (PAR) reaching the plant top was ensured and maintained at the level of 500–600 μmol m$^2$ s$^{-1}$. Clay pebbles substrate can also provide a sufficient biofiltration, increasing the efficiency of the system.

### 2.3. Abiotic Factors

At the beginning of the experiment and after an initial period of 24 h that permitted any trace of chlorine to escape, two grains of a previously conditioned salt water aquarium's filter bed were introduced to each aquaponic system serving as inoculums for nitrification bacteria [29]. To start up the aquaponic systems, 0.2 g $NH_4Cl$ was added and dissolved in each system to be used as an ammonia source [30]. Total ammonia nitrogen (TAN), nitrite and nitrate ions, pH, dissolved oxygen and salinity were monitored every two days and approximately less than 5% of the water in each system was replaced every day with water of the tested salinities (8 ppt and 20 ppt). Total ammonia nitrogen (TAN), nitrite and nitrate ions were measured using a HACH 3800 model photometer with special pre-weighted reagents for each of the nitrogen compounds. The pH and the dissolved oxygen were also monitored electronically (HACH HQ 40D).

Furthermore, total organic carbon (TOC) and total nitrogen (TN) were analysed using a Shimadzu TOC analyser (TOC-VCSH) coupled with a chemilumine scence detector (TNM-1 TN unit), thus creating a simultaneous analysis system. TOC analysis was performed using the Combustion-Infrared method [31]. The principle of this method is that a microportion of the sample is injected into a heated reaction chamber packed with an oxidative catalyst, which in our case was $Pt/Al_2O_3$. The organic and inorganic carbon is then oxidized to $CO_2$ and water. The $CO_2$ is transported in the carrier gas streams (purified air) and is measured by means of a non-dispersive infrared analyzer (NDIR analyzer). Because in this way Total Carbon (TC) is estimated, Inorganic Carbon (IC) is measured separately, by acidifying the sample with HCl acid at pH < 3. TOC is obtained by difference (TOC = TC−IC). TN analysis was performed using the Pyrolysis-Chemilumine scence detection method, where oxidative pyrolysis converts chemically bound nitrogen to nitric oxide (NO) which is contacted with ozone ($O_3$) producing metastable nitrogen dioxide ($NO_2$*). As the $NO_2$* decays, light is emitted and detected by a photomultiplier tube.

Bacterial abundance was monitored every two weeks by flurochrome staining [30]. To measure the abundance of the bacteria, 20 mL of water was taken from each aquaponic system on day (d) 0, 15 d, 30 d, 45 d, 60 d and 72 d. The sample was fixed with 2% formaldehyde final concentration and was kept at 4 °C in the dark. A sub sample of 10–15 mL was filtered on black Nuclepore filters (pore size of 0.2 μm) and stained with DAPI (4′,6-diamidino-2-phenylindole). After mounting the filters on glass slides, the cells were counted on an Axiostar (Zeiss) epifluorescence microscope at ×1000 magnification [32]. DAPI counts were counted three times for every sampling time point from the same tank and the coefficient of variation was always less than 10%. Bacteria abundance was computed according to equations previously described [30].

To control the efficiency and the function of the filter bed, thehydraulic loading ratio (HLR), recycled ratio (r), the hydraulic retention time of the water in the filter bed (HRT), the specific surface

area of the filter (SSA), the volume of filter media ($V_{media}$) and the filter volume (V) were calculated according to equations previously described [7,16]. HLR (m/day) = flow rate (Q)/total surface area of the trough, where:

- HRT (min) = (surface area water x depth x porosity of gravel trough/flow rate)
- SSA ($m^2/m^3$) = Surface area of filter media/volume of the filter media
- $V_{media}$ ($m^3$) = surface area of the filter media/SSA
- r = volume of recycled water/volume of the system

*2.4. Biological Analyses*

At the end of the trial, the following indexes were calculated [33], where $W_{in}$ and $W_{fin}$ are the initial and final weight of the fish respectively, and $\Delta t$ is the duration of the experiment in days:

- SGR (%/day) = (($\ln W_{fin} - \ln W_{in}$)/$\Delta t$) $\times$ 100
- WG (gr) = $W_{fin} - W_{in}$
- FCR = Food offered (gr)/weight gain (gr)
- DFI (%/d) = 100 $\times$ ((food offered/weight gain)/feeding days)
- Condition factor (K) = (W $\times$ $L^{-3}$) $\times$ 100 where W is the body weight of the fish (gr) and L is the total length of the fish (cm)
- S = ((final number of fish − initial number of fish)/initial number of fish) $\times$ 100

The growth rate of the plants was calculated according to equation [19].

- Plant Growth (cm/d) = height of plant/day

Plants' growth performance was monitored every two weeks by measuring the plant height of all individuals per treatment on a 1.1 $m^2$ hydroponic trough area, by counting the number of shoots and the number of lateral branches of the plants. At the final harvest, plant growth characteristics were measured in terms of root and aerial part biomass. The two plant parts were separated and oven-dried at 70 °C, until constant weight was obtained.

At the end of the experiment, five fish were taken from each tank for histopathological examination. Fish were euthanized with increased dosage of MS-222 and were placed immediately on ice. Samples of gills, liver, kidney and midgut were taken from each fish. Tissue samples were first fixed in Davidson' fixative for 24 h at 4 °C and then immediately dehydrated in graded series of ethanol, were immersed in xylol, and finally embedded in paraffin wax. Sections of 5–10 μm were mounted. After the samples have been deparaffinized, the sections were rehydrated, then stained with Hematoxylin-Eosin and mounted with Cristal/Mount and were examined for alterations with a microscope (Axiostar plus Carl Zeiss Light Microscopy, Carl Zeiss Ltd., Gottingen, Germany) under a total magnification of 100× and 400×. A semi-quantitative grading system was used in order to quantify the histopathological alterations of the examined tissues [34]. The severity grading used the following system: Grade 0 (not remarkable), Grade 1 (minimal), Grade 2 (mild), Grade 3 (moderate), Grade 4 (severe).

At the end of the experiment, samples of gills and intestine were excised from six (6) randomly selected fish taken from each tank and they were frozen in liquid nitrogen and stored at −80 °C until analyzed. SDS-PAGE (sodium dodecyl sulfate–polyacrylamide gel electrophoresis) and immunoblot analysis was performed as previously described [35]. Briefly, frozen tissues were homogenized in 3 mL $g^{-1}$ of cold lysis buffer based on 20 mM β-glycerophosphate, 50 mM NaF, 2 mM EDTA (ethylenediaminetetraacetic acid), 20 mM Hepes (4-(2-hydroxyethyl)-1-piperazine ethanesulfonic acid), 0.2 mM $Na_3VO_4$, 10 mM benzamidine, pH 7, containing 200 μM leupeptin, 10 μM trans-epoxy succinyl-L-leucylamido-(4-guanidino) butane, 5 mM DTT (dithiothreitol), 300 μM phenyl methyl sulfonyl fluoride (PMSF) and 1% *v/v* Triton X-100, and extracted on ice for 30 min. Thereafter, samples were centrifuged at 10,000 *g*, for 10 min in 4 °C. The supernatants were boiled with 0.33 volumes of SDS/PAGE sample buffer containing 330 mM Tris-HCl, 13% *v/v* glycerol, 133 mM DTT, 10% *w/v* Sodium

Dodecyl Sulfate and 0.2% *w/v* bromophenol blue. Protein concentration was determined using the BioRad protein assay, a dye-binding assay based on the differential colour change of a dye (Coomansie Brilliant Blue G-250) in response to various protein concentrations.

Equivalent amounts of protein (50 μg) were separated on 10% (*w/v*) acrylamide, 0.275% (*w/v*) bisacrylamide slab gels and were transferred electrophoretically onto nitrocellulose membranes (0.45 μm, Schleicher and Schuell, Keene, NH 03431, USA). Non-specific binding sites on the membranes were blocked with 5% (*w/v*) non-fat milk in TBST (Tris Buffered Saline-Twin 20) (20 mM Tris-HCl, pH 7.5, 137 mM NaCl, 0.1% (*v/v*) Tween 20) for 30 min at room temperature. Thereafter, the membranes were incubated overnight with the appropriate primary antibodies. The antibodies used were as follows: monoclonal rabbit anti-heat shock protein (D6F1) XP, 60 kDa (Cat. No. 12165, Cell Signaling, Beverly, MA, USA), monoclonal rabbit anti-heat shock protein, 90 kDa (Cat. No. 4874, Cell Signaling, Beverly, MA, USA), monoclonal rabbit anti-phospho p 44/42 MAPK (Erk1/2) (Thr202/Tyr204) (D13.14.4E) XP (Cat. No. 4370, Cell Signaling, Beverly, MA, USA), monoclonal rabbit anti-phospho-p38 MAPK (Thr180-Tyr182) (Cat.No. 9211, Cell Signaling, Beverly, MA, USA), monoclonal rabbit anti-phospho-SAPK/JNK (c-Jun N-terminal kinases), (Thr183/Tyr185) (81E11), (Cat. No. 4668, Cell Signaling, Beverly, MA, USA). After washing in TBST ($3 \times 5$ min), the blots were incubated with horseradish peroxidase-linked secondary antibodies, polyclonal goat anti-rabbit immunoglobulins (Cat. No. 7074, Cell Signaling, Beverly, MA, USA), were washed again in TBST ($3 \times 5$ min), and the bands were detected by enhanced chemilumine scence (Cell Signaling, Beverly, MA, USA) with exposure to Fuji Medical X-ray films. The films were quantified by Image Studio Lite (Quantification Software, LI-COR Biosciences).

### 2.5. Statistical Analysis

Values are presented as means ± standard error (S.E.M). Data were tested for normality and homogeneity with Kolmogorov–Smirnov and Levene's tests, respectively. Independent t-tests were considered statistically significant at $p < 0.05$ [31]. Statistical analyses were carried out using the software package IBM SPSS Statistics V22.

## 3. Results

### 3.1. Abiotic Factors

There were no significant differences (t-test, $p > 0.05$) in the means of TAN, nitrite, nitrate ions, and pH concerning the water quality in both treatments (8 ppt and 20 ppt aquaponic system), (Table 1). Figure 1, presents the variation of the above parameters during the study period.Values of TAN concentration fluctuated from 0.39 ± 0.1 mg/L to 0.33 ± 0.8 mg/L for the 8 ppt and 20 ppt salinity treatment, respectively. TAN has reached its minimum value (<0.05 mg/L) on day 75 of the experiment (Figure 1). Nitrite concentrations were in the range of 0.89 ± 0.3 mg/L and 0.82 ± 0.3 mg/L for the 8 ppt and 20 ppt salinities, respectively, while the values were decreased on day 35 of the experiment (Figure 1). The mean value of the nitrate ions fluctuated from 76.4 ± 11.2 mg/L to 77.2 ± 11.8 mg/L for the 8 ppt and 20 ppt salinities, respectively. The nitrate in both aquaponic systems showed an increasing tendency from day five until the end of the experiment (Figure 1).

**Table 1.** The water quality of the brackish aquaponic system developed under two different salinities (8 ppt and 20 ppt) during the study period (102 days).

|  | 8 ppt | 20 ppt |
|---|---|---|
| TAN (mg/L) | 0.39 ± 0.1 [a] | 0.33 ± 0.07 [a] |
| $NO_2^-$ (mg/L) | 0.89 ± 0.3 [a] | 0.82 ± 0.3 [a] |
| $NO_3^-$ (mg/L) | 76.4 ± 11.2 [a] | 77.2 ± 11.8 [a] |
| pH | 7.54 ± 0.05 [a] | 7.73 ± 0.042 [a] |

Data are expressed as mean ± S.E.M (n = 22). Means in a row followed by the same superscript are not significantly different ($p > 0.05$). TAN: total ammonium nitrogen, $NO_2^-$: nitrite ions, $NO_3^-$: nitrate ions.

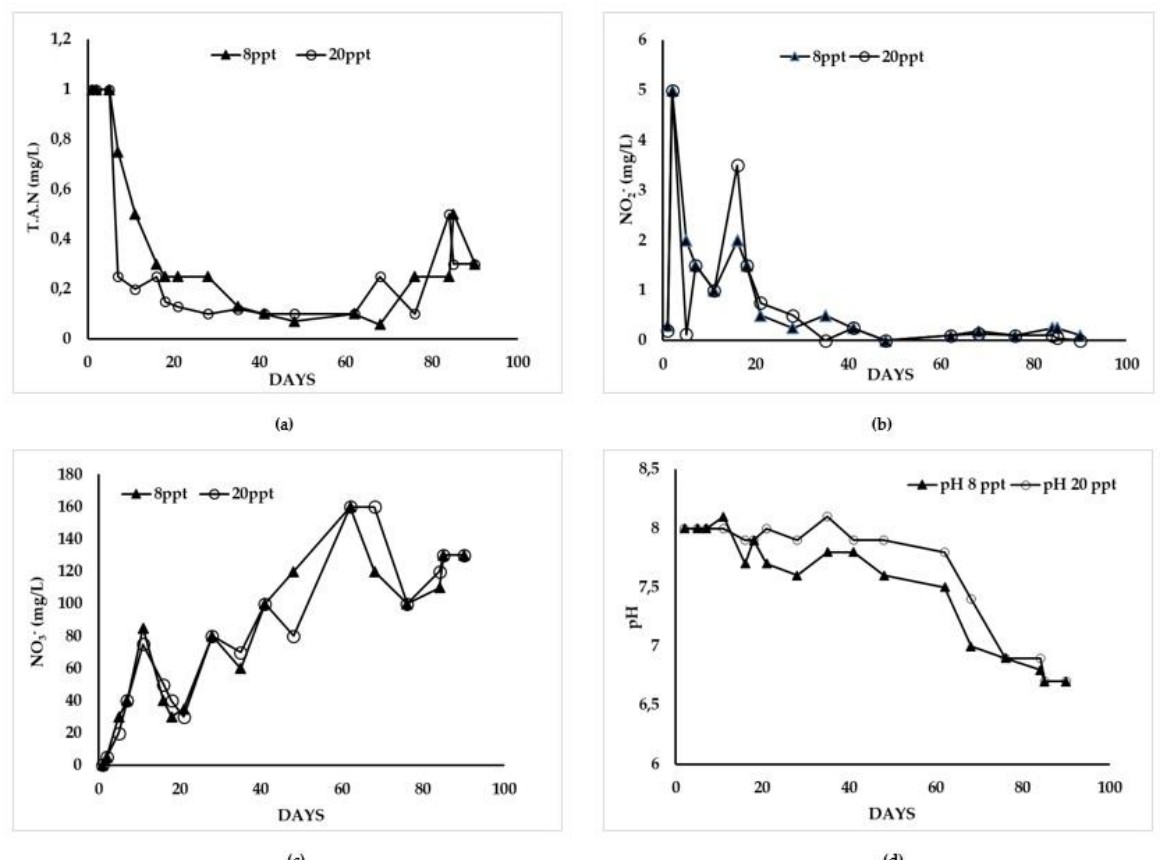

**Figure 1.** T.A.N (**a**), nitrite (**b**), nitrate concentration (**c**) and pH (**d**) during the study period.

The results showed that the mean values of TN and TOC were significantly higher at 20 ppt than 8 ppt treatment (t-test, $p < 0.05$) (Table 2). Moreover, no significant differences at the C/N ratio were observed between the treatments (t-test, $p < 0.05$) (Table 2).

**Table 2.** (Mean ± S.E.M) concentration of inorganic and organic nutrients during the experiment period of 102 days in the brackish aquaponic system developed in this study.

|  | 20 ppt | 8 ppt |
|---|---|---|
| TN (mg/L) | 23.89 ± 6.30 [a] | 14.42 ± 7.77 [b] |
| TOC (mg/L) | 16.57 ± 2.0 [a] | 8.1 ± 0.78 [b] |
| C/N | 1.14 ± 0.32 [a] | 0.66 ± 0.1 [a] |

Data are shown as mean ± S.E.M.(n = 8). Means in a row followed by the same superscript are not significantly different ($p < 0.05$). TN: total nitrogen, TOC: total organic carbon.

At the end of the experiment (102 days), the mean final concentrations of TN at 20 ppt ($23.89 \pm 6.30$ mg/L) was significantly higher compared to 8 ppt ($14.42 \pm 7.77$ mg/L) (t-test, $p < 0.05$) (Table 2). Moreover in 20 ppt salinity, TOC showed a statistically higher concentration ($16.57 \pm 2.0$ mg/L) in comparison to the 8 ppt ($8.1 \pm 0.78$ mg/L) salinity treatment (t-test, $p < 0.05$) (Table 2). TN showed the highest concentration on day 74 on both treatments (Figure 2a). Concentration of TOC gradually increased during the experimental period, reaching a value of 26.46 mg/L and 12.28 mg/L for 20 ppt and 8 ppt salinity respectively, on day 102 of the experiment (Figure 2b).

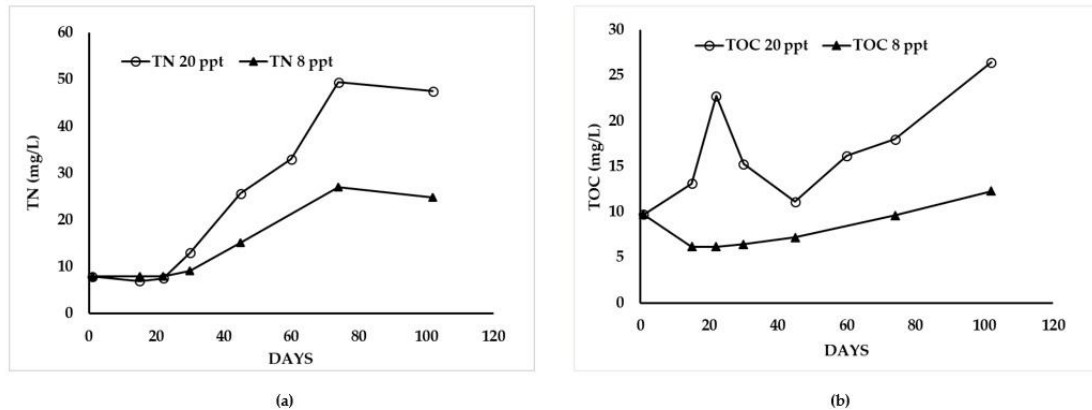

**Figure 2.** TN (**a**) and TOC (**b**) concentrations measured over the experimental period of 102 days.

The C/N ratio throughout the experiment fluctuated from $1.14 \pm 0.32$ for the salinity of 20 ppt and $0.66 \pm 0.10$ for the 8 ppt salinity, without showing statistically significant differences between the treatments (t-test, $p > 0.05$), (Table 2). Figure 3, presents the values of the C/N ratio (carbon:nitrogen) and TN concentration for both salinities. In both salinities the increase of TN concentration was followed by a reduction in C/N ratio. In the salinity of the 20 ppt the C/N ratio took values above 1 for the main part of the experiment (Figure 3a), while for the 8 ppt salinity the values ranged from 0.49 to 1 (Figure 3b).

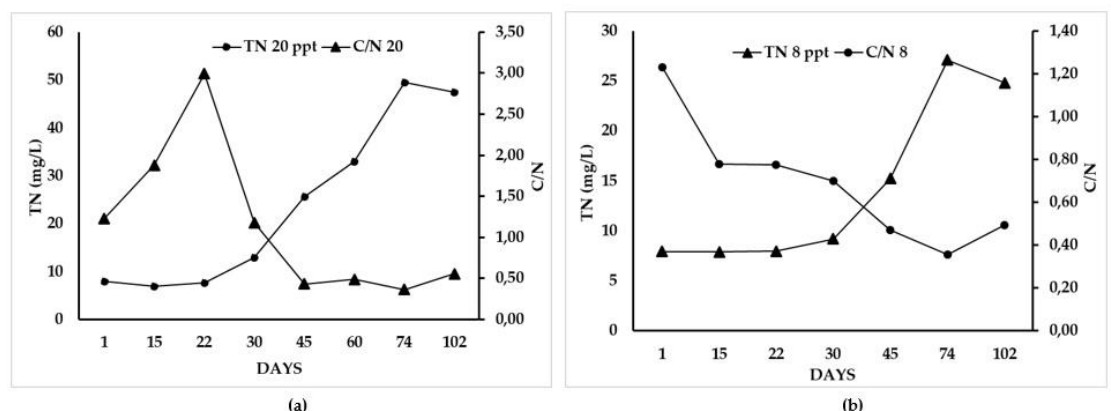

**Figure 3.** TN concentration and C/N ratio in both brackish aquaponics systems 20 ppt (**a**) and 8 ppt (**b**) during the experiment (102 days).

There were no significant differences in the means of the hydraulic loading rate (HLR), recirculation rate (r), hydraulic retention time (HRT) and filter's volume (V) (t-test, $p > 0.05$) for the 8 ppt and 20 ppt salinity treatments, respectively (Table 3). The specific surface area of the filter bed (SSA) and the $V_{media}$ was significant higher in 20 ppt compared to 8 ppt in which it was lower (t-test, $p < 0.05$), indicating higher ammonia removal rate in the filter.

**Table 3.** Functional characteristics of the filter in both brackish aquaponic system (8 ppt and 20 ppt) during the study period.

|  | 8 ppt | 20 ppt |
|---|---|---|
| HLR (m/day) | $1.85 \pm 0.005$ [a] | $1.85 \pm 0.005$ [a] |
| R (min) | $0.014 \pm 0.003$ [a] | $0.014 \pm 0.003$ [a] |
| HRT (min) | $9.7 \pm 0.01$ [a] | $9.7 \pm 0.02$ [a] |
| SSA ($m^2/m^3$) | $162 \pm 0.30$ [a] | $224 \pm 0.31$ [b] |
| $V_{media}$ ($m^3$) | $6.73 \pm 0.02$ [a] | $6.94 \pm 0.08$ [b] |
| V (L) | $32.1 \pm 0.001$ [a] | $32.1 \pm 0.001$ [a] |

Data are expressed as mean $\pm$ S.E.M (n = 6). Means in a row followed by the same superscript are not significantly different ($p > 0.05$). HLR: hydraulic loading rate, r: recirculation rate, HRT: hydraulic retention time, SSA: specific surface area, $V_{media}$: volume of the filter media, V: volume of the filter.

### 3.2. Biotic Factors

The fish and plant growth performances are illustrated in Table 4. At the start of the experiment, there were no significant differences in the means of the juvenile sea breams' initial body weight (gr) and length (cm), (t-test, $p > 0.05$) for both salinity treatments (Table 4). There were no statistically significant differences in the means of the final body weight (g), final length (cm), specific growth rate (%/day), or weight gain (g) in both treatments at the end of the experiment (t-test, $p > 0.05$). The survival rate for gilthead juvenile sea bream over 102 days of culture was 99% and 97% in 8 ppt and 20 ppt salinity treatments, respectively. The initial condition factor ($K_{in}$) was significantly higher (t-test, $p < 0.05$) in 20 ppt ($1.46 \pm 0.02$) compared to 8 ppt ($1.35 \pm 0.02$). The final condition factor ($K_{fin}$) of the gilthead juvenile sea bream during the experimental period was significantly higher (t-test, $p < 0.05$) in 8 ppt in comparison to 20 ppt ($1.46 \pm 0.02$). There were no significantly differences (t-test, $p > 0.05$) in FCR and daily food intake (DFI) indexes between the gilthead juvenile sea bream cultured in both 8 ppt and 20 ppt salinity treatments (Table 4) and values ranged from $1.80 \pm 0.06$ (8 ppt) to $1.4 \pm 0.07$ (20 ppt) and $2.39 \pm 0.08$ (8 ppt) to $2.45 \pm 0.1$ (20 ppt), respectively.

**Table 4.** Fish and plant growth performances in the brackish aquaponics system under two different salinities during the study period.

|  | 8 ppt | 20 ppt |
|---|---|---|
| **Gilthead seabream growth performance** | | |
| Initial weight ($W_{in}$, g) | $2.54 \pm 0.05$ [a] | $2.55 \pm 0.06$ [a] |
| Final weight ($W_{fin}$, g) | $27.91 \pm 0.84$ [a] | $28.07 \pm 0.85$ [a] |
| Weight gain (WG, g) | $25.36 \pm 0.83$ [a] | $25.51 \pm 0.87$ [a] |
| Specific growth rate (SGR, %/day) | $3.17 \pm 0.04$ [a] | $3.17 \pm 0.06$ [a] |
| Survival (%) | 99% | 97% |
| Initial condition factor ($K_{in}$) | $1.35 \pm 0.02$ [a] | $1.46 \pm 0.02$ [b] |
| Final condition factor ($K_{fin}$) | $1.58 \pm 0.03$ [a] | $1.46 \pm 0.02$ [b] |
| Food Conversion rate (FCR) | $1.80 \pm 0.06$ [a] | $1.84 \pm 0.07$ [b] |
| Daily Feed Intake (DFI, %/day) | $2.39 \pm 0.08$ [a] | $2.45 \pm 0.1$ [a] |
| Initial length ($L_{in}$, cm) | $5.73 \pm 0.03$ [a] | $5.57 \pm 0.04$ [a] |
| Final length ($L_{fin}$, cm) | $12.04 \pm 0.11$ [a] | $12.40 \pm 0.16$ [a] |
| **Rock samphire growth performance** | | |
| Initial height (cm) | $8.45 \pm 0.34$ [a] | $7.94 \pm 0.46$ [a] |
| Final height (cm) | $10.32 \pm 0.58$ [a] | $8.03 \pm 0.47$ [b] |
| Height gain of plant | $1.87 \pm 0.56$ [a] | $0.99 \pm 0.28$ [b] |
| Final number of lateral branches | $4.83 \pm 0.47$ [a] | $2.94 \pm 0.42$ [b] |
| Final number of shoots | $4.22 \pm 0.59$ [a] | $1.94 \pm 0.21$ [b] |

Data are expressed as means $\pm$ S.E.M ($n_1$ = 156 gilthead seabream fish and $n_2$ = 36 rock samphire plant individuals). Means in a row followed by the same superscript are not significantly different ($p > 0.05$).

Regarding plant growth performance, at the start of the experiment there were no significant differences between the initial heights (cm) of rock samphire (t-test, $p > 0.05$) in both treatments. During the study period, no mortality or plant disease was observed for rock samphire in both brackish aquaponics systems. The final mean height of the plant (cm) was significantly higher in the aquaponic system with the 8 ppt salinity (10.32 ± 0.58 cm), compared to the aquaponic system in which the salinity was 20 ppt (8.03 ± 0.47 cm). At the end of the experiment, rock samphire had a significantly higher final number of shoots and additional lateral branches at 8 ppt salinity compared to the 20 ppt salinity (t-test, $p < 0.05$) (Table 4). Concerning final biomass, plants grown under 8 ppt accumulated significantly more biomass in both the aerial part and roots, compared to 20 ppt (Figure 4).

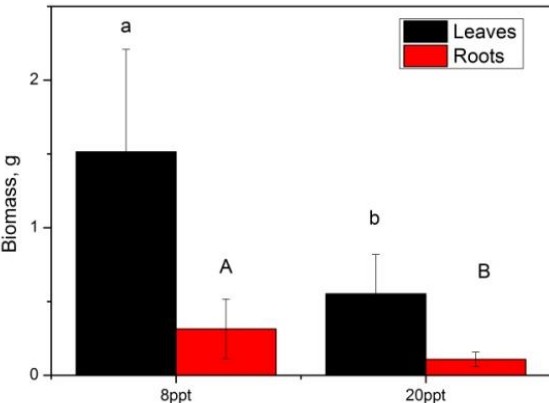

**Figure 4.** Plant biomass of leaves and roots of rock samphire in both treatments during the experimental process. Different letters denote statistically significant differences at $p < 0.05$ (small letters for leaves and capital letters for roots).

The midgut microscopic examination showed histopathological alterations in only one tank of the 20 ppt salinity group, with minimal (Grade 1) lipid accumulation in the enterocytes (Table 5, Figure 5). Liver histopathology of the 20 ppt salinity group revealed only minimal (Grade 1) accumulation of lipid droplets in liver cells (Table 5, Figure 5). One fish of this group showed mild liver haemorrhage. Liver histopathology of the 8 ppt salinity group revealed mild histopathological alterations, with areas of necrosis, inflammation, haemmoradge and steatosis (Table 5).

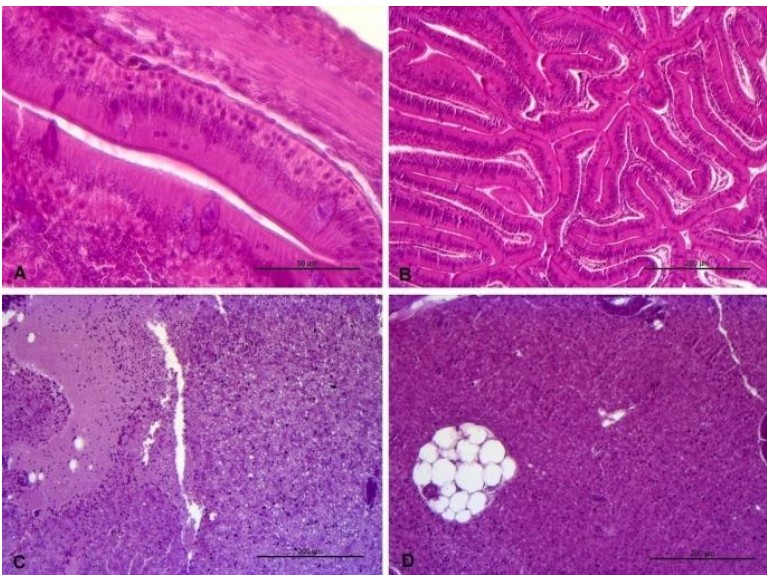

**Figure 5.** (**A**) 8 ppt salinity group. Normal intestine. (**B**) 20 ppt salinity group. Normal intestine. (**C**) 8 ppt salinity group. Liver with necrotic area. (**D**) 20 ppt salinity group. Liver with steatosis.

**Table 5.** Severity score (0–4) for the observed histopathological alterations of gilthead juvenile sea bream.

|  | 8 ppt | 20 ppt |
|---|---|---|
| Midgut | 0 | 0 |
| Liver | 2 | 1 |
| Kidney | 3 | 2 |
| Gills | 2 | 2 |

The juvenile sea breams kept in 20 ppt salinity showed larger collecting ducts in their kidney than the ones in 8 ppt salinity (Figure 6). There were mild and moderate histopathological alterations for the 20 ppt salinity group and for the 8 ppt salinity group, respectively (Table 5). The most important were: Granulomas, haemorrhage, necrosis, glomerular hyperplasia/hypoplasia, dilation of Bowman space, hyperplasia of tubules' wall, steatosis, and loss of tubules' lumen. Mild histopathological alterations were detected in both groups (Table 5) in the gills. The most important were: hyperplasia of primary lamellae, hyperplasia of secondary lamellae and epithelium detachment at the secondary lamella.

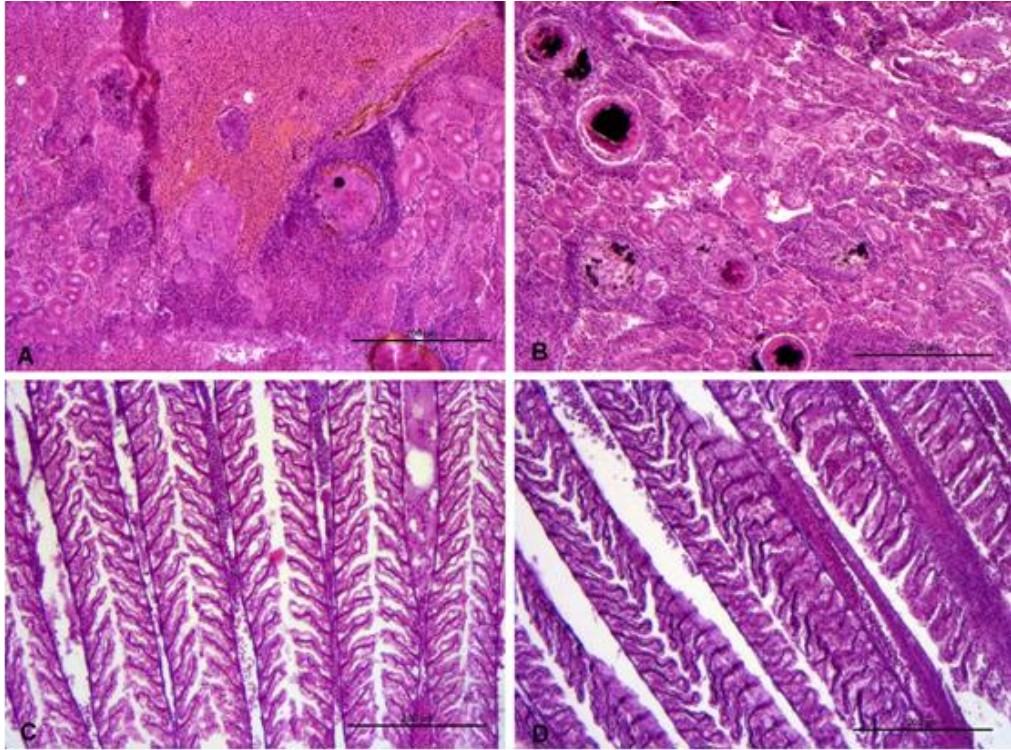

**Figure 6.** (**A**) 8 ppt salinity group. Kidney with haemmoradge, granuloma and glomerular hypertrophy. Melanomacrophages centers are also present. (**B**) 20 ppt salinity group. Kidney with many granulomas. Melanomacrophages centers are also present. (**C**) 8 ppt salinity group. Hyperplasia of primary, hyperplasia of secondary lamellae and epithelium detachment at the secondary lamella. (**D**) 20 ppt salinity group. Hyperplasia of primary and secondary lamellae.

The HSP levels were not significantly affected in the two examined tissues of the gilthead seabreams exposed to different salinities (Figure 7). However, in the intestine, HSP60 (Figure 7A) levels were significantly higher in fish derived from the 8 ppt salinity system compared to fish from 20 ppt.

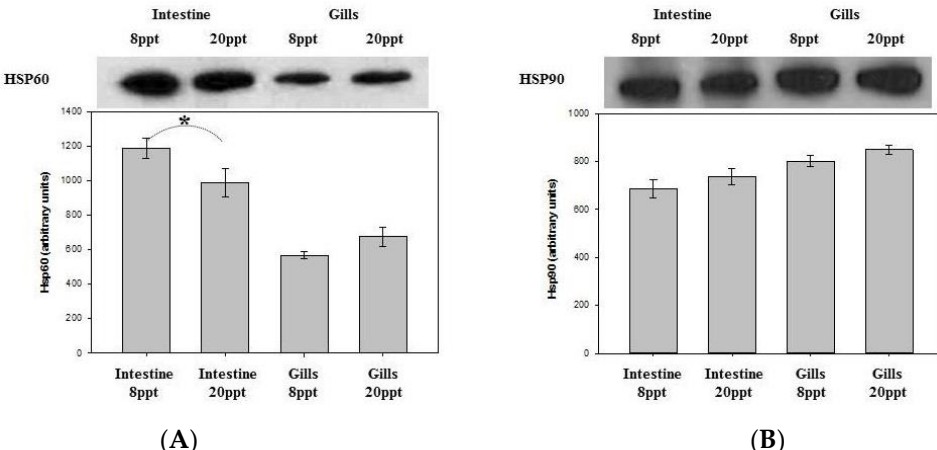

**Figure 7.** HSP60 (**A**) and HSP90 (**B**) levels in the intestine and gills of *Sparus aurata* at 8 and 20 ppt salinities. Values represent mean ± standard deviation (SD) of 6 determinations; asterisk (*) represents significant difference between the two examined salinities, $p < 0.05$.

The phosphorylation levels of MAPKs (p38 MAPK, p44/42 MAPK) and JNK (c-Jun N-terminal kinases) in intestine and gills of the gilthead seabream examined in both salinities are illustrated in Figure 8. Similar profiles of the three examined MAPKs are found in both tissues sampled from the different salinities. Specifically, MAPK phosphorylation levels were higher in both tissues from 20 ppt compared to those form 8 ppt. All the levels of the studied MAPKs showed significant differences between the two examined salinities, with the exception of phospho JNK in the intestine.

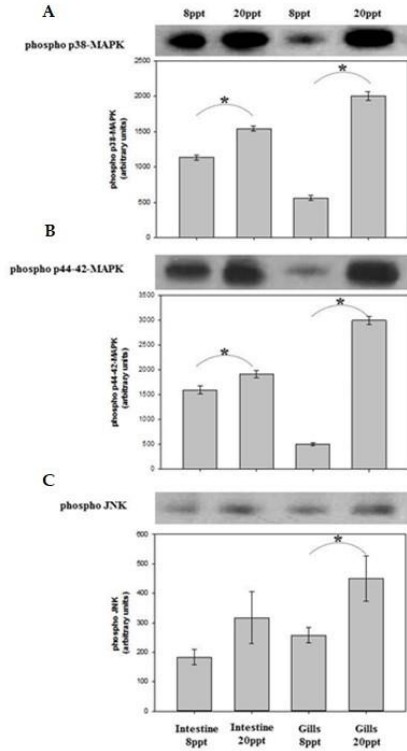

**Figure 8.** Phosphorylation levels of p38 mitogen-activated protein kinase (MAPK) (**A**), p44/42 MAPK (**B**) and c-Jun N-terminal kinases (JNKs)(**C**) in the intestine and gills of *Sparus aurata* at 8 and 20 ppt salinities. Values represent mean ± SD of 6 determinations; asterisk (*) represents significant difference between the two examined salinities, $p < 0.05$.

The bacterial abundance at the beginning of the experiment and before the change in the salinity to 20 and 8 ppt, showed no statistically significant differences between the two treatments (t-test, $p > 0.05$) (Table 6). The results showed that the abundance of bacteria at the start of the experiment, when the salinity was set to 20 ppt and 8 ppt, in all treatments was statistically higher ($20.2 \times 10^5 \pm 0.92$ cells/mL) at 20 ppt in comparison to the lower salinity (8 ppt) where the abundance of cells was $1.08 \times 10^5 \pm 1.82$ cells/mL (Table 6). The average abundance of bacteria at the end of the experiment was higher (t-test, $p < 0.05$) in the 20 ppt salinity ($18.6 \pm 0.91$ cells $\times 10^5$/mL) than the 8 ppt salinity ($6.8 \pm 1.9$ cells $\times 10^5$/mL) (Table 6). The pH values in all treatments were between 8 to 6.7 and there was a reduction of bacteria abundance at the pH of 6.7 (Figure 9).

**Table 6.** Bacterial abundance of the brackish aquaponic system developed under two different salinities (8 ppt and 20 ppt) during the study period.

|  | 8 ppt | 20 ppt |
|---|---|---|
| Initial bacterial abundance (cells $\times 10^5$/mL) | $19.9 \times 10^5 \pm 0.72$ [a] | $20.2 \times 10^5 \pm 0.92$ [a] |
| Bacterial abundance (cells $\times 10^5$/mL) at the 1st day of salinity changes | $1.08 \times 10^5 \pm 1.82$ [a] | $20.2 \times 10^5 \pm 0.92$ [b] |
| Final bacterial abundance (cells $\times 10^5$/mL) | $6.8 \times 10^5 \pm 1.9$ [a] | $18.6 \times 10^5 \pm 0.91$ [b] |

Data are expressed as means ± S.E.M (n = 10). Means in a row followed by the same superscript are not significantly different ($p > 0.05$).

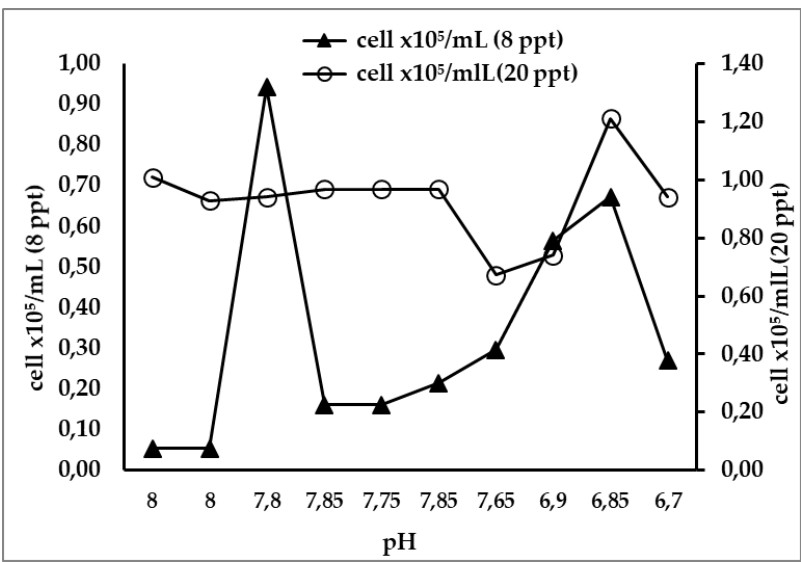

**Figure 9.** Bacterial abundance and variation of pH in the biofilter of the brackish water aquaponics system developed under two different salinities 8 and 20 ppt during the study period.

## 4. Discussion

### 4.1. Abiotic Environment

In the present study an experimental brackish aquaponic system for Mediterranean fish (gilthead juvenile seabream) and halophytes (rock samphire) was investigated for a duration of 102 days, and to authors' knowledge, this is the first such investigation. To date, only a few studies have been conducted on the use of brackish water in aquaponic systems [5,21,24], whereas the use of freshwater in aquaponic systems has been widely studied [5,21,24,36]. A successful aquaponic system provides important benefits, such as water quality control, appropriate fish and plant growth performances, plant and fish disease management, eliminating environmental impacts and functioning as an RAS. In this way, such systems do not require large volumes of fresh water daily (less than 5% renewed water is needed due to evaporation or losses from daily functioning) [37,38].

Gilthead seabream is one of the most valuable commercial fish species of the Mediterranean, and it has shown very good adaptability to aquaculture [39,40]. The present study showed that the use of brackish water in aquaponics is efficient since the conditioning of the system is associated with the biological filter and achieved by minimizing ammonia and nitrite concentrations and increasing nitrate concentrations. The results of the study suggest that the nitrate concentration could maintain lower TAN concentrations during the experimental period and indicate that an equal amount of TAN oxidation to nitrate occurred for both treatments (8 ppt and 20 ppt). These results were suitable for rock samphire absorption and growth performance, which is consistent with previously reported results [21,24,25], thus indicating that similar fluctuations in nitrate ions and pH were maintained within the recommended safety limits [3]. The conditioning of the filter at the RAS occurred via the effectiveness of the filter bed, which reduced the ammonia and nitrite concentrations and produced higher levels of nitrates, which is similar to the values reported by Spotte [6].

Previous studies [30,33] reported that the conditioning of the freshwater aquariums of two ornamental fish (cichlid zebra and angel fish) was achieved over the short term via biochemical nitrate production due to an increase in the abundance of nitrifying bacteria. The maintenance of water quality parameters is essential to ensuring the best aquaculture conditions for optimal fish performance and avoiding adverse conditions which could affect fish's optimal growth, survival and health in an RAS system [9,10,41–43].

The hydraulic retention time (HRT) of the present study for both salinities of the aquaponic system was 9.7 min/day, indicating that the biofilter performed efficiently (nitrification and denitrification rate) and removed ammonia. Timmons and Ebeling [44] reported that in a saltwater RAS, an HRT of 2 h affected the denitrification process. In the same study, an increase of the HRT to 6 h decreased the denitrification rate. The HRT has an impact on the carrying capacity of the filter and especially affects the ammonia removal efficiency [45,46], alkalinity production [47], sulphate production [48] and C/N ratio in the denitrification process [49]. Furthermore, the hydraulic loading rate (HLR) has an impact on fish and plant production and nutrient removal and is correlated with the daily food input in the system for the optimum plant ratio [25,50]. Chen et al. [50] suggested that the best HLR for a freshwater aquaponic system is 1.28 m/day because it provides the best production performance for fish (SGR: 1.80%/day) and plant growth (1.75 cm/day). In the present study, the HLR (1.85 m/day) for both treatments was even higher, thus providing a higher growth performance for fish (SGR: 3.17%/day) and plants (PGR: 1.87 cm/day for 20 ppt and 0.99 cm/day for 8 ppt).

Concerning the C/N ratio, it has been reported to affect the oxidation capacity of the filter bed and depends on the daily feed supply and nutrient input to the system [11]. Hence, a balance is required between bacterial abundance and fish and plant performance [3]. Zhu and Chen [51] reported that the load can affect the process of nitrification and the relationship with heterotrophic bacteria and autotrophic nitrifying bacteria in an aquaponic system. Additionally, according to Ohashi et al. [52], the efficiency of the filter decreases when the C/N ratio increases, thus leading to a higher abundance of heterotrophic bacteria, and depends on the TAN concentration. High C/N ratios of heterotrophic bacteria reduce the diffusion of nitrogen and dissolved oxygen (DO) in autotrophic nitrifying bacteria, thereby adversely affecting the rate of nitrification [53,54]. Moreover, Zhu and Chen [51] found that when the C/N ratio is less than 1, the biological process of nitrification is affected.

In this study, at the 20 ppt salinity, the C/N ratio was above 1, while at the 8 ppt salinity, the C/N ranged from 0.49 to 1 (Figure 3b). Furthermore, at both salinities (20 and 8 ppt), an increase of the TN concentration was followed by a reduction in the C/N ratio. In the 20 ppt salinity treatment, the C/N ratio was 1.14 and the carrying capacity of the filter bed appeared to be affected. Additionally, the oxidation capacity of the filter was affected by the abundance of heterotrophic bacteria because of the increase in the total nitrogen concentration in the system, although the plant roots as well as the medium increased the surface for nitrification by ensuring the removal of TAN. Reports have shown that when the C/N varies from 1.5 to 2.5, TAN and nitrate removal is necessary for their absorption by the plants [6]. Timmons and Ebeling [40] also reported that the C/N ratio at 2.3 seems to ensure

the optimal rate of denitrification by increasing the TN reduction rates, whereas in the present study, plants contributed to the reduction of nitrates via adsorption, which lead to a subsequent increase in plant height. Spotte [6] has suggested that to initiate the process and set up a closed system, the C/N value should be close to 1. According to the results of the present study, the C/N ratio at 20 ppt (Figure 3) increased from week one to week three and appeared to be inversely proportional to the reduction of TN. On the other hand, the C/N ratio at 8 ppt decreased from the first week to the end of the experiment (Figure 3) and appeared to be inversely proportional to an increase in TN, which was probably because of the reduced nitrification rate associated with an increase in heterotrophic bacteria. Other studies by Gullian and Aramburu and Zhu and Chen [41,51] found that the increased C/N ratio was not inversely proportional to a reduction in the nitrification process due to the available DO in the RAS, although the C/N ratio was greater than 2.7 after the third week of experimentation.

In addition, in this study, the bacterial abundance was measured in two brackish aquaponic systems during the start-up period of system conditioning by generating constant conditions, such as daily feed input rates and daily ammonia production rates. Before the salinity was changed to 20 and 8 ppt, the abundance of bacteria did not show statistically significant differences ($p > 0.05$). The results showed that the abundance of bacteria at the start of the experiment (when the salinity was set to the two requested salinities 20 ppt and 8 ppt) in all treatments was statistically higher ($20.2 \times 10^5 \pm 0.92$ cells/mL) at 20 ppt compared with 8ppt, where the abundance of cells was $1.08 \times 10^5 \pm 1.82$ cells/mL. The average abundance of bacteria at the end of the experiment was higher ($p < 0.05$) at 20 ppt salinity ($18.6 \pm 0.91$ cells $\times 10^5$/mL) than at 8 ppt salinity ($6.8 \pm 1.9$ cells $\times 10^5$/mL). The decreased bacterial abundance observed at 8 ppt in the present study may be related to salinity differences between both treatments and different C/N ratios and HLR, HRT and pH values [3,11,55]. Previous studies [30] reported that the bacterial abundance in two ornamental freshwater aquariums reached $29 \times 10^5$ cells/mL and $12 \times 10^5$ cells/mL for zebra cichlid fish and angelfish, respectively. In batch (closed) cultures, the increase in bacterial cell numbers is usually attributed to a few dominant species, although this was not the case in our study. The bacterial abundance continued to increase until the end of the experiment due to the lack of considerable grazing pressure [56].

Previous studies conducted in RASs have shown similar increases in the bacterial abundance tendency relative to the results presented in this study and presented values below $0.005 \times 10^5$ cells/mL at RAS start up to $0.2 \times 10^5$ cells/mL at the end of the experiment [11,57,58]. Somerville et al. [3] suggested that in a freshwater aquaponic system, the system balance depends on the nitrate concentration, daily feed input, HLR, SSA of the filter, pH and bacterial abundance. Other studies [59] suggested that the HRT in the RAS affects the preservation and maturation of bacteria in the filter. As the HRT of the water in the filter increases, bacterial maturation is more easily maintained, which leads to an increase in bacterial abundance. Alleman et al. [60] reported that nitrifying bacteria exhibit increased resistance to salinity changes and short adaptation times. The bacteria shrink to salinities higher than 2 M [61] due to electrostatic contraction of the cell and osmotic reactions.

## 4.2. Biological Elements

The results of the present study show that gilthead juvenile sea bream in a brackish aquaponic system presented a high growth performance (SGR = 3.17%/day), survival rate, and FCR. Khater et al. [21] reported that tilapia cultured in a brackish aquaponic system showed a high survival rate and increased its weight by 0.78 g/fish/day for four months. Additionally, Pantanella and Colla [24] and Waller et al. [62] suggested that sea bass grew on average from 32 g to 54 g at an SGR of 1.5%/day and FCR of 0.93 in a 16 psu salinity aquaponic system over 35 days. Nozi et al. [25] studied the quality of seabass (*Dicentrarchus labrax*) in a brackish aquaponic system (continuous salinity changes from 35 ppt to 0 ppt) to determine the optimal plant growth performance via the addition of inorganic nutrients, such as iron, potassium and calcium and found that the fish were affected by the conditions created. The same study [25] reported that the levels of saturated and polyunsaturated fatty acids were not similar at all salinity adjustments.

Rock samphire was chosen due to its high nutritional value, aromatic-pharmaceutical properties and natural tolerance to high salt concentrations [27]. The results of the present study showed that rock samphire showed statistically significant better growth in terms of biomass, height and number of lateral branches at 8 ppt salinity than at 20 ppt salinity. Rock samphire is a facultative halophyte, meaning that it can grow in a wide range of salinity levels. While it thrives when the irrigation water contains up to 250 mM NaCl, it exhibits reduced growth rates when the salinity exceeds this limit [28]. Nevertheless, rock samphire can survive in seawater [63]. Previous studies indicated that the range of optimal performance in a hydroponics system occurs between 35 and 171 mM NaCl [28], which corresponds to approximately 1.8–10 ppt. This finding is consistent with our results, where the optimal growth of rock samphire occurred at 8 ppt salinity. Nozi et al. [25] cultivated *Beta vulgaris varcicla* in a brackish water aquaponic system and found that it presented a statistically significant increase in the growth and length of roots. In addition, Kotzen et al. [22] reported that eggplant, mint, celery and chili pepper were successfully grown in a brackish aquaponic system, whereas spinach, chives and tomato presented poor yields [21,22].

In addition, the two examined tissues of the gilthead seabream exposed to different salinities exhibit a differential HSP induction and MAPK activation with the esxeption of phosphor JNK in the intestine. The critical role and the significance of the HSPs expression and activation of MAPKs in relation to physiological functions when juvenile sea breams are exposed in different salinities remains to be investigated. In euryhaline fish, such as juvenile sea breams, the kidney plays an important role in osmoregulation. Sea bass, trout, herring, and juvenile sea bream can adapt to changes in salinity and are able to survive in both seawater and freshwater. According to Nebel et al. [64], sea bass juveniles that were successfully adapted to freshwater showed smaller collecting ducts than those cultivated in seawater, which is consistent with our results. The kidney granulomas that were detected in both salinity groups are likely correlated with the long-term storage of formulated feeds or with ascorbic acid deficiency [65,66]. The absence or presence of mild histopathological alterations in the midgut, gills and liver of both groups (20 and 8 ppt) indicates that the fish were well adapted to freshwater. Similar results were reported by Giffard-Mena et al. [67], Laiz-Carrión et al. [68], and Masroor et al. [69], thus indicating the high plasticity and gill remodelling of sea bass adapted from seawater to freshwater.

## 5. Conclusions

This study advances our understanding of aquaponic systems by establishing an effective productive system using brackish water as an alternative source of water, which represents a remarkable innovation because it can contribute to the economic development of regions and countries that have an increased demand for saltwater fish and abundance of seawater or brackish resources but limited direct access to fresh water. Aquaponic systems contribute positively to increased food production and food security.

This study highlights that the simultaneous cultivation of seawater fish and plants (halophytes) with high nutritional and increased commercial value in brackish water aquaponic systems is feasible and sustainable. Moreover, this study creates a comparison method and provides new research data, which can be used as a reference for future research. The development of brackish or seawater aquaponic systems can help to reduce the use of freshwater resources for food production since such systems are implemented in a controlled environment, thus allowing for the reuse of water, reduction of waste disposal to the environment and minimization of marine pollution. In addition, brackish or seawater aquaponic systems represent a form of organic farming with high standards for safe food production. However, additional research into aquaponic systems is needed to further promote the production of high-quality food with high commercial and nutritional value in the future.

**Author Contributions:** Conceptualization N.V., methodology, N.V., K.K., E.A., E.L., N.K., V.B., P.B.; formal analysis, N.V., E.M., K.K., E.A., P.B., N.K., V.B. and P.S.; data curation, N.V., K.K., E.A., V.B. and P.B.; writing—original draft preparation, N.V. and E.M.; writing—review and editing, N.V., E.M., E.A., K.K., N.K., V.B., E.L. and P.B.; supervision, N.V., E.M.

**Funding:** This research received no external funding.

**Acknowledgments:** The authors wish to express their thanks to E. Patsea, S. Frangou, I. Mitsopoulos, K. Babouklis, E. Kapetanios, A. Demetriou, S. Paschos and K. Morfesis for their help with the start up and the growth experiment. In addition, the authors are very grateful to SELONDA SA and BIOMAR for the generous sponsorship of the gilthead seabream fish and feed used in the experiment, respectively.The rock samphire (*Crithmum maritimum*) plant material that was used in the experiment originated from Mount Athos, Greece (International Plant Exchange Netwrok Code GR-1-BBGK-16.5961) and was propagated by the Institute of Plant Breeding and Genetic Resources, Hellenic Agricultural Organization Demeter.

**Conflicts of Interest:** The authors declare no conflict of interest.

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
