# Peer review of "An Experimental Brackish Aquaponic System Using Juvenile Gilthead Sea Bream (Sparus aurata) and Rock Samphire (Crithmum maritimum)"

_sustainability, doi:10.3390/su11184820_

Round 1
Reviewer 1 Report
This is an interesting paper that provides production information along with physiology. While the experiment was fairly small, it provides some good initial research into brackish water aquaponics. What this paper really needs is a thorough edit to clean up punctuation and grammatical errors. Once that is covered I think this is acceptable for publication.
Author Response
Reviewer 1
This is an interesting paper that provides production information along with physiology. While the experiment was fairly small, it provides some good initial research into brackish water aquaponics. What this paper really needs is a thorough edit to clean up punctuation and grammatical errors. Once that is covered I think this is acceptable for publication.
The paper was read and corrected by a native English speaker.

Reviewer 2 Report
This is a great piece of work and well structured. However, it has been spoiled by very obvious grammar errors. In many instances there are no spaces between words. Please check this - it is appearing all the way from introduction to the conclusion. I know English may not be your first language. There is need for English editing of some words that are misspelled eg page 2 line 59 "provifying" instead of "purifying"; page 14 line 461 "fileter" instead of filter. On the methodology please explain briefly why you chose only two treatments of 8ppt and 20ppt All figures in the results should be revised - the Y and X axis figuers are so blurred There are some areas where there are problems of citing eg page 14 line 462 - .... values reported by (6) please complete by who? and then provide your reference. Similary line 477 - .... it was suggested by (16) etcAuthor Response
Reviewer 2
This is a great piece of work and well structured. However, it has been spoiled by very obvious grammar errors. In many instances there are no spaces between words. Please check this - it is appearing all the way from introduction to the conclusion. I know English may not be your first language. There is need for English editing of some words that are misspelled eg page 2 line 59 "provifying" instead of "purifying"; page 14 line 461 "fileter" instead of filter. On the methodology please explain briefly why you chose only two treatments of 8ppt and 20ppt All figures in the results should be revised - the Y and X axis figuers are so blurred There are some areas where there are problems of citing eg page 14 line 462 - .... values reported by (6) please complete by who? and then provide your reference. Similary line 477 - .... it was suggested by (16) etc
We corrected all the spaces between words and misspelled words and revised the English in the manuscript. We explain at the methodology section why we selected the two salinities (Lines 114-118).
All figures in the results and citations are revised and all citations are numbered correctly. All references in the text are corrected according to the reviewer’ comments and marked in yellow.

Reviewer 3 Report
in 338 row is number of table 3 should be 4.
General comment - work done very well and carefully but the question arises how this translates into practice - FCR = 1.8 is not promising.
Author Response
Reviewer 3
in 338 row is number of table 3 should be 4.
Corrected according to the reviewer’ comment.
General comment - work done very well and carefully but the question arises how this translates into practice - FCR = 1.8 is not promising.
The FCR is indeed high but this is the first study to investigate a combined production of fish and plants in a brackish aquaponics system using gilthead sea bream and rock samphire. Further research will investigate all the factors and parameters which are limiting for fish and plant production in a recirculated aquaponics system with aim to improve and optimize the FCR.
